# BAG-LEVEL SELF-SUPERVISED INSTANCE BASED DISTANCE IN MULTIPLE INSTANCE LEARNING

## ABSTRACT

Multiple Instance Learning (MIL) methods are typically supervised. However, a bag-to-bag metric is needed in many applications, including clustering, statistical tests, and dimension reduction. Such a metric should differentiate between bags, regardless of the sparsity or overlap between the instances of the bags. We propose SUMIT (Self sUpervised MIL dIsTance) as an instance-embedding-based distance that maximizes the distinction between bags. SUMIT is optimized using five criteria: self-similarity within a bag, quality of instance reconstruction, robustness to sampling depth, conservation of triangle inequality, and separation of instances to clusters. We show using current standard MIL datasets and a novel wiki-based set of wiki topics that the within bag-similarity loss is the most important for a bag-to-bag metric that best separates bags of similar classes. SUMIT bridges the gap between instance-level and bag-level approaches, by keeping the embedding of all instances but ensuring their proximity within a bag.

## 1 INTRODUCTION

Multiple Instance Learning (MIL) is weakly supervised learning where the training instances are not stand-alone examples, but instead are arranged in sets, called bags(2). Labels are provided for the entire bag and not for the individual samples, further denoted as instances. To exemplify MIL, consider chapters in a book. The sentences can be considered as the individual instance, and the chapter is a bag. In the training set, we know that chapters belong to a category. The goal of MIL classification is to detect whether a chapter in the book belongs to a category. While a large number of algorithms were developed for MIL classification, there is no inherent method to define distances between bags. Note that many methods were proposed to define distances on sets (e.g. the energy distance(20), Wasserstein Distance(21) Hausdorff distance(27), etc). However, these distances do not consider the distribution of instances in each bag. We propose to combine self-supervised learning and metric learning to produce such a metric between bags. Using the book chapters example above- when considering the chapters as the bags, one would expect the distances between chapters in the same book to be smaller than the distance between chapters from other books. Also, the distances between chapters from books of the same genre or subject should be smaller than the distances between chapters from other genres or subjects.

MIL classification methods can be grouped into three categories: instance-space (IS), bag-space (BS), and embedded-space (ES) methods(30). The categorization is based on how information from the bag is extracted. The Instance-Space (IS) paradigm is based on local, instance-level information, ignoring the characteristics of the whole bag. This is based on the assumption that the information lies in the instances. The Bag-Space (BS) paradigm is based on global, bag-level information, and each bag is treated as an entity. The Embedded-Space (ES) paradigm, like the BS paradigm, is based on global, bag-information. In contrast with the BS paradigm, which uses a non-vectorial representation of the bag, in the ES paradigm, each bag is mapped to a single feature vector.

We here propose a combination of the instance and bag level, by computing a distance on an embedding of each instance, requiring that the distance between instances of the same bag to be closer than instances between bags. We do not assume any label on the bag. As such, the metric presented here is an instance-level self-supervised distance at the bag level. This approach combines the advantages of instance and bag-level approaches.

We deviate from the mainly supervised approach used in MIL. A well-defined metric in the MIL setting has multiple benefits, including, distance-based machine learning (e.g. KNN(11)), the definition of density in the bag space using kernel density estimation - KDE(7), clustering using distance-based clustering (e.g. community detection(16) or dBscan (8), dimension reduction using distance conserving projection (e.g. MDS (15),UMAP (17), t-SNE(25) and the resulting visualization when projected to two or three dimensions, and distance based statistical tests(3).

## 2    RELATED WORK

While no method was explicitly developed for a self-supervised MIL metric definition, some supervised and unsupervised methods provide a distance metric between bags. In general, one can produce a distance from all ES and BS representation methods.

1. **ES (Embedding Space)** methods in MIL explicitly map entire bags into continuously embeddings(13; 29). Instance embeddings can be combined using pooling(10) or attention(26) to compute bag-level metrics. The Euclidean (or any other) distance between the embedding of the bags is a metric.

2. **BS (Bag Similarity)** (27; 9; 33; 6; 23; 32; 12; 28) methods indirectly derive distances between bags by first computing distances or similarities between instances within bags, often using kernel functions or distance metrics like the Hausdorff distance.

3. **IS (Instance Space)** methods consider instances individually and aggregate their predictions or features to infer bag-level decisions. There are no bag-to-bag good Instance-based metrics. In the last decade, there has been much less interest in instance-based methods, since those suffer from the aggregation problem - how to combine the instances to represent a bag. Each combination method has limitations. Using the mean can lead to instances canceling one another, and using each instance creates different size representations for each bag (note that the same problem may emerge in ES).

Other methods applied existing distances to specific problems (Appendix A). The main works that explicitly produced distances are:

1. Citation-KNN(27) proposed a modified Hausdorff Distance-based MIL approach. The Hausdorff distance is utilized to provide a metric function between subsets of a metric space. Specifically, two bags $X,Y$ are in distance $d$ if every instance of bag $X$ is within distance d of at least one of the instances in bag $Y$:

$$d_{Hausdorff}(X_i, X_j) = max(h(X_i, X_j), h(X_j, X_i)); h(X_i, X_j) = \max_{x_i \in X_i} \min_{x_j \in X_j} ||x_i - x_j||$$
(1)

2. Kernel Methods(9) define a kernel function as the sum of instance kernels:

$$d_{Kernel}(X_i, X_j) = K(X_i, X_j) := \sum_{x_i \in X_i, x_j \in X_j} k(x_i, x_j)$$
(2)

3. Graph Kernel(33) transforms the bags into undirected graphs implicitly or explicitly, and applies standard distance metrics of graphs such as edit-distance(22) and inexact matching distance(4):

$$d_{\text{Graph}}(X_i, X_j) = D(G_{X_i}, G_{X_j})$$
(3)

,where $G_{X_i}$ is the graph produced from $X_i$.

4. MInd(6) represents each bag by a vector of its dissimilarities to other bags (e.g. Hausdorff distance) in the training set and treats these dissimilarities as a feature representation:

$$d_{MInd}(X_i, X_j) = d(F_{MInd}(X_i), F_{MInd}(X_j)), \quad F_{MInd}(X_i) = [d(B_i, B_1), \ldots, d(B_i, B_M)]$$
(4)

where the bag $B_i$ is compared to each of $B_j, 1 <= j <= M$ the M bags of a training set.

5. Contrastive Multiple Instance Learning(23): Method initially trains a tile-wise encoder using SimCLR(5), from which subsets of tile-wise (instances) embeddings are extracted and fused via an attention-based multiple-instance learning framework to yield slide-level (bag) representations. The resulting set of intra-slide-level and inter-slide-level embeddings are attracted and repelled via contrastive loss, respectively.

6. Multi-instance clustering with applications to multi-instance prediction(32): Each bag is represented by a k-dimensional feature vector, where the value of the i-th feature is the distance between the bag and the medoid (the center). The medoid is set in the minimum distance $medoid = \underset{X_i \in G}{\arg\min} \sum_{X_j \in G} d(X_i, X_j)$ to all bags in group $G$, where $d$ is the distance to the medoid) of the i-th group. The distance between bags can be Min, Max, or Hausdorff of the instances.

# 3 NOVELTY

All current metrics use either a projection of the entire bag and as such lose most of the information on the bag, or use the original representation of each instance combined with some predefined distance between sets. This suffers from a high overlap between bags.

Here we propose a novel approach - use an energy distance via the embedding of each instance. The embedding of each instance is learned to ensure that the distance between instances of different bags is larger than the distance of instances in a bag. In addition, we optimize the embedding of each instance to ensure that the distance between bags adheres to the requirements of a distance metric (e.g. the triangle inequality.

Specifically, we do not optimize the distances between bags. Instead, we optimize the embedding of each instance to minimize five different losses:

1. **Reconstruction Loss.** We minimize the discrepancy between the original and the reconstructed features following an encoder-decoder model.
2. **Contrastive Loss** We minimize the distance between instances of the same bag compared to distances from different bags.
3. **Invariance Loss.** We ensure the distance is not sensitive to the number of instances sampled in each bag.
4. **Clustering.** We optimize the division of the encoding into distinct clusters.
5. **Triangle Loss.** We maximize the gap between the undirect distance from one instance to another through a third instance and the direct distance between said instances (adherence to the triangular inequality theorem).

To the best of our knowledge, this is the first time the embedding of each instance is optimized to produce an optimal distance between bags.

# 4 METHODS

## 4.1 DATA

We used two types of datasets1: benchmark MIL datasets(18)(1)(24) and a self-curated dataset from Wikipedia pages (Appendix B for the list of topics). The wiki data is available in the github.

In the benchmark datasets, the MUSK1 and MUSK2 datasets represent molecules identified as musks or non-musks by human experts. The features of each instance represent the shape of the molecules, and the different instances are the different conformations of the molecules. The Fox, Tiger, and Elephant datasets are subsets of the Corel image retrieval dataset, separated into images presenting (or not) the associated animal. Each image is a bag, and the instances are created by dividing the image into segments, where each segment's features represent the segment's color, texture, and shape. If any image segment includes the animal, the bag is labeled positive. The Wiki dataset introduced here is a novel dataset developed for the current analysis. The bags are Wikipedia pages of cities in different countries, such that each country is a label of the city. The instances are the Bag Of Words (BOW) of each section on the page.

## 4.2 MODEL AND EXPERIMENTAL SETUP

We used an encoder-decoder model, which was trained using the losses below. Let $x \in \mathbb{R}^{d_{\text{input}}}$ represent the input data, where $d_{\text{input}}$ is the input instances dimension. We used an encoder-

| Data Sets | | | | |
|-----------|-----------|---------------|----------|------------------|
| Name | No. Features | No. Instances | No. Bags | Data Type |
| MUSK1 | 166 | 476 | 92 | Tabular |
| MUSK2 | 166 | 6598 | 102 | Tabular |
| FOX | 230 | 1320 | 200 | Tabular |
| TIGER | 230 | 1220 | 200 | Tabular |
| ELEPHANT | 230 | 1391 | 200 | Tabular |
| Wikipedia | 1205 | 19568 | 1504 | Text Sequences |

Table 1: Data sets statistics- the number of different features (the instance dimension), the total number of instances (samples), and bags, and the type of data.

decoder model with layers: $[d_{\text{input}}, 128, 64, 32, 64, 128, d_{\text{input}}]$, applying ReLU activations and Batch Normalization(14) between each layer. The encoding is $\tilde{x}_i$. The output of the decoder is $\hat{x}_i$.

The latent layer is normalized using Layer Normalization (LP). For the Wikipedia dataset, the encoder-decoder has layers: $[d_{\text{input}}, 16, 10, 16, d_{\text{input}}]$. The model was trained using 5-fold cross-validation with an 80-20 train-test split, over 2,000 epochs, using the Adadelta optimizer(31).

### 4.3 LOSS FUNCTIONS

We tested five loss functions1 aimed at optimizing the reconstruction, but also ensuring a high distance between bags.

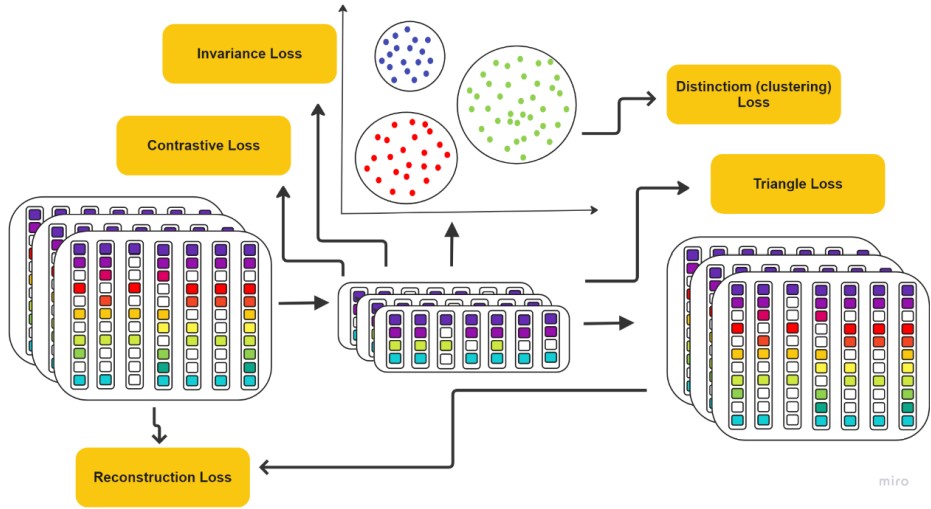

Figure 1: The five losses schema. The scheme shows the bags (rightmost large stack) that are made of the instances (rectangles inside squares). The instances are projected through the model to a latent space (middle stack) and then projected back to the original dimension (left-most stack). The different losses are derived from different layers of the model. **Reconstruction** from the difference between the original and reconstructed bags, **Distinction** from the latent space (through clustering, as illustrated in the three centroids on the middle-top axis), and then **Contrastive**, **Invariance** from the latent space at the bag level, and **Triangle** at the instance level.

- **Reconstruction Loss** The norm of the distance between the decoder output and the original input for each instance:

$$L_{Reconstruction} = \|x_i - \hat{x}_i\|_2 \tag{5}$$

- **Contrastive Loss.** Minimization of the distance between pairs of instances originating from the same bag and maximizing the distance between pairs from different bags.

$$L_{Contrastive} = -log\frac{\exp\left(\frac{sim(\tilde{x_i},\tilde{x_j})}{\tau}\right)}{\sum_{k=1}^{2N}\exp\left(\frac{sim(\tilde{x_i},\tilde{x_k})}{\tau}\right)}, where \ \tilde{x_i},\tilde{x_j} \in X_\alpha \ and \ \tilde{x_k} \notin X_\alpha \quad (6)$$

- **Invariance Loss**. The invariance loss ensures the distance is not sensitive to the sample depth, defined as the number of instances sampled from each bag.

$$L_{Invariance} = -log\frac{\exp\left(\frac{sim(Agg(\tilde{X_i}),Agg(\tilde{X_j}))}{\tau}\right)}{\sum_{k=1}^{2N}\exp\left(\frac{sim(Agg(\tilde{X_i}),Agg(\tilde{X_k}))}{\tau}\right)},$$

$$where \ Agg(\tilde{X_i}), Agg(\tilde{X_j}) \ are \ an \ aggreggation \ of \ the \ the \ same \ bag \ X_\alpha \ in$$

$$different \ depths \ and \ Agg(\tilde{X_k}) \ of \ other \ bag \ X_\beta. \quad (7)$$

- **Clustering Loss.** The clustering loss is tailored to maximize the clustering of the encoded instances, akin to the objectives pursued in the K-Means algorithm. Specifically, it aims to minimize the within-cluster variances, measured as squared Euclidean distances:

$$L_{Clustering} = \frac{SSE}{\sum_i \sum_j \|x_i - x_j\|} = \sum_{i=1}^{K} \sum_{x_j \in C_i} \frac{\|c_i - x_j\|^2}{\sum_i \sum_j \|x_i - x_j\|} \quad (8)$$

The clustering is done using the method proposed by Xai beyond classification(19) where the data is encoded and then clustered, where the clustering is optimized every other epoch.

- **Triangle Loss.** The triangle loss is designed to optimize the adherence to the triangular inequality principle. The distances between points forming the vertices of a triangle are represented by the Euclidean distances between the corresponding instance projections. It is computed as the percentage of deviation from the triangular inequality across all possible triples of data points. It is defined as follows:

$$L_{Triangular} = p(d(x_i, x_j) > d(x_i, x_k) + d(x_k, x_j)) \quad (9)$$

This loss tries to spread the points in wide angles.

## 4.4 ENERGY DISTANCE

We use the energy distance(20) for the bag-to-bag distance. Each bag is represented as a set of its instances projection in $\mathbb{R}^n$. For two bags $X$. $Y$, it is:

$$D(X,Y) = 2\mathbb{E}[\|X - Y\|] - \mathbb{E}[\|X - X'\|] - \mathbb{E}[\|Y - Y'\|]. \quad (10)$$

### 4.4.1 KERNEL DENSITY ESTIMATION

To ensure that the projection maximizes the similarity of instances within the same bag, we use a kernel-density estimate (KDE) (Eq. 11). KDE is a non-parametric method used to estimate the probability density function of a random variable in a metric space. As seen in the schematic figure 2, it smooths the data points using a kernel function, providing a non-parametric continuous estimate of the underlying distribution. Here we use a class-dependent KDE. For each instance of class A, we compute the density of instances of either class A (self) or class B (other), using a Gaussian kernel with a radius $h$.

$$\hat{f}(X) = \frac{1}{nh}\sum_{i=1}^{n} K\left(\frac{d(X.X_i)}{h}\right); K(d) \qquad = \frac{1}{\sqrt{2\pi}}\exp\left(-\frac{d^2}{2}\right) \quad (11)$$

$h$ was defined as $2 * \sigma(X)$ - the standard deviation of all the distances between the bags. We define the quality of the embedding using the ratio of the self to other densities:

$$S = \frac{Average_X KDE(Self, X)}{Average_X KDE(Other, X)} \quad (12)$$

## 5 RESULTS

We propose here to learn an embedding of the instances, and then apply an energy distance to the bags. The embedding is learned using an encoder-decoder architecture with five different losses:

1. Reconstruction: error between the original input features and the reconstructed features produced by a mode

2. Contrastive: minimize distance for pairs from the same bag, maximize distance for pairs from different bags.

3. Invariance: maintains consistent distances between the same pair of bags across different sampling depths.

4. Clustering: Minimizes the bag's distance to assigned clusters, aiming at creating a more dense representation.

5. Triangle: optimizes adherence to the triangular inequality by minimizing deviations in Euclidean distances between points forming a triangle.

To evaluate the quality of the distance, we compared the density of bags from class A or B around bags from class $A$ (as estimated by a KDE). A good projection should maximize the ratio of the self (density of A around A) to other (density of B around A) densities.

To illustrate that, we generated toy data of bags from two labels, where the instances of bags with one label are far from the instances of bags with the second label. Following the method above, the bag-to-bag distance was calculated using energy distance, and the bag projections were visualized using multi-dimensional scaling (MDS). (Fig. 2 A). The higher KDE for bags from the same label (A, A) vs bags of different labels (A, B) captures the separation of the bags as seen in (Fig. 2 b). We propose the same approach to estimate the quality of the distance on real-world datasets.

### 5.1 LIMITATION OF INSTANCE-BASED DISTANCES

When we applied the same approach to real-world datasets (using pre-defined projections of the instances), a poor separation was noticed between bags of similar labels and those of different labels, as can be observed by the low densities, as well as similar density distributions between the self and other densities (Fig. 2 d). One can see that for example the MDS projection based on the bag distance (computed similarly to the toy model) of the Elephant dataset (Fig. 2 d), reveals a lack of visual distinctiveness among labels, with the red and blue points overlapping. This observation is further supported by the low ratio between the KDE values for the distances based on instances across all datasets (Fig. 2 b).

### 5.2 AN AUTOENCODER THAT LEARNS THE INSTANCE DISTRIBUTION DOES NOT IMPROVE THE BAG DISTANCE METRIC.

If indeed instances are closer within a bag than between bags, an embedding of the instances learned via an auto-encoder (AE) could in theory ensure a more uniform distribution of the instances within bags, and as such improve the within-to-between bag density. We trained an AE using only a reconstruction loss (i.e. only ensuring that the AE reproduces the input). However, in general, the AE only decreases the within-to-between class density (Fig 3C). The reconstruction loss decreases to very low values, suggesting a good reconstruction of the instance values. However, using the encoded latent representation to compute the bag-based distance does not improve the within-to-between class density ratio.

### 5.3 THE WITHIN-TO-BETWEEN BAG DENSITY RATIO CAN BE IMPROVED USING OTHER LOSSES

Reconstructing the original instance space does not improve the within-to-between class density ratio. We thus tested alternative losses. Those include Contrastive, Invariance, and Triangle losses. When training with one of the losses at a time, each of the four losses converges rapidly (See musk data results in Fig. 3 a. The results are similar for other datasets). The ratio between the density of each of the labels improves and grows over time for the contrastive and invariance losses, but not

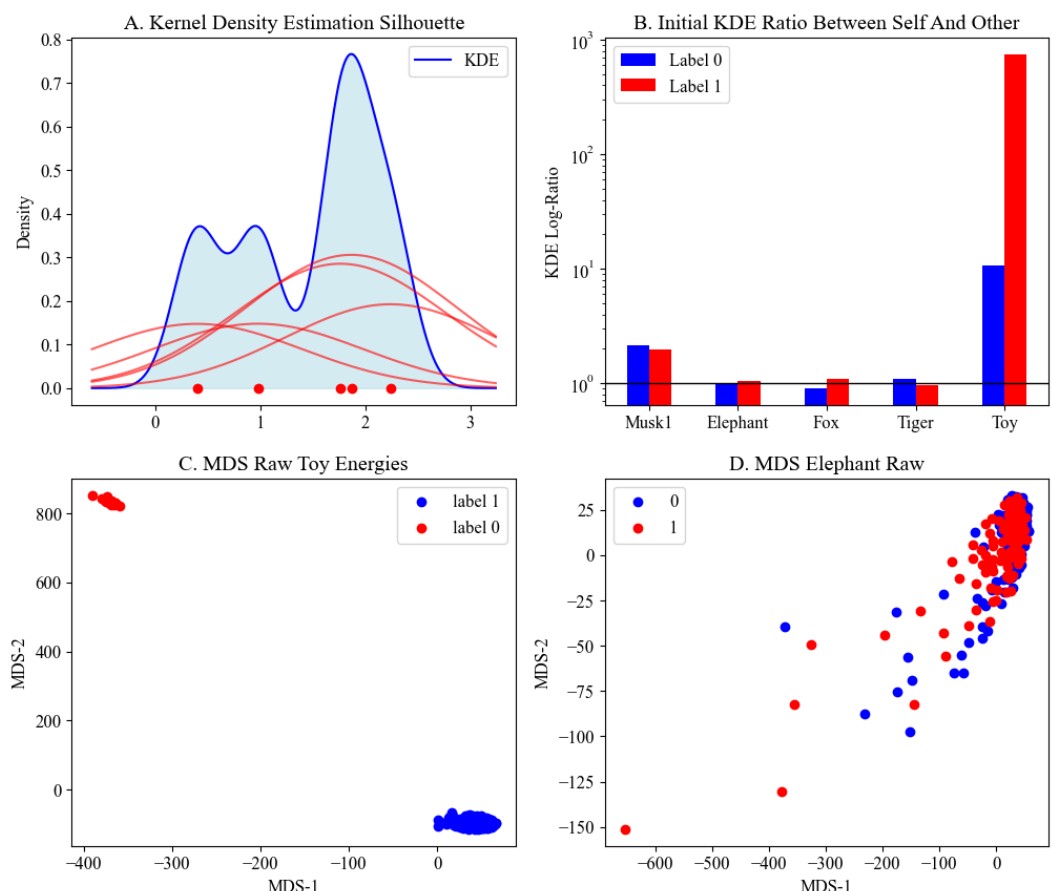

Figure 2: **(a)** KDE schematic showing smoothing of data points. **(c)** MDS visualization for toy data and raw Elephant data **(d)**. Axes represent MDS dimensions. Blue and red denote labels 0 and 1. Label 0 of toy data in range [0, 2] and label 1 in range [10,15]. Toy data shows clear label separation, while instance-based distance on Elephant lacks distinct separation. This correlates with the high KDE values for Toy data and low for Elephant as seen **(b)** in bar plot for KDE of both labels.

for the reconstruction or triangle loss (See Fig. 3b for Musk - results are similar for other datasets). Indeed, following convergence, both Invariance and contrastive losses increase the within to between class densities, in contrast with the reconstruction and triangle that reduce the ratio (See average for training and validation in Fig. 3c, and distribution over 5 folds in Fig. 3d).

## 5.4 MULTI-CLASS CLASSIFICATION USING NOVEL WIKIPEDIA DATASET

To test that the conclusions above can be extended to multiple classes, we developed a new MIL dataset composed of the BOW of Wikipedia cities descriptions from different countries (See Fig. 4 d for the number of cities per country). The class of each bag was its country. We computed the density of the bag own country bags vs other countries at the end of training. (Fig. 4b for average relative improvement per country in within to between class density in the validation set, and 4c. for the distribution). Again, the contrastive and invariance losses induce the best improvement consistently over all cities. the other losses have practically no effect.

## 5.5 INCLUSION OF CLUSTERING LOSS AND COMBINATION OF LOSSES

A simple solution to increase the self-density is to produce a projection that would better cluster. We added a clustering loss that mimics a k-means clustering (see methods). For each loss, (reconstruction, contrastive, invariance, and triangle) we trained the model with the loss fixed to 1 and

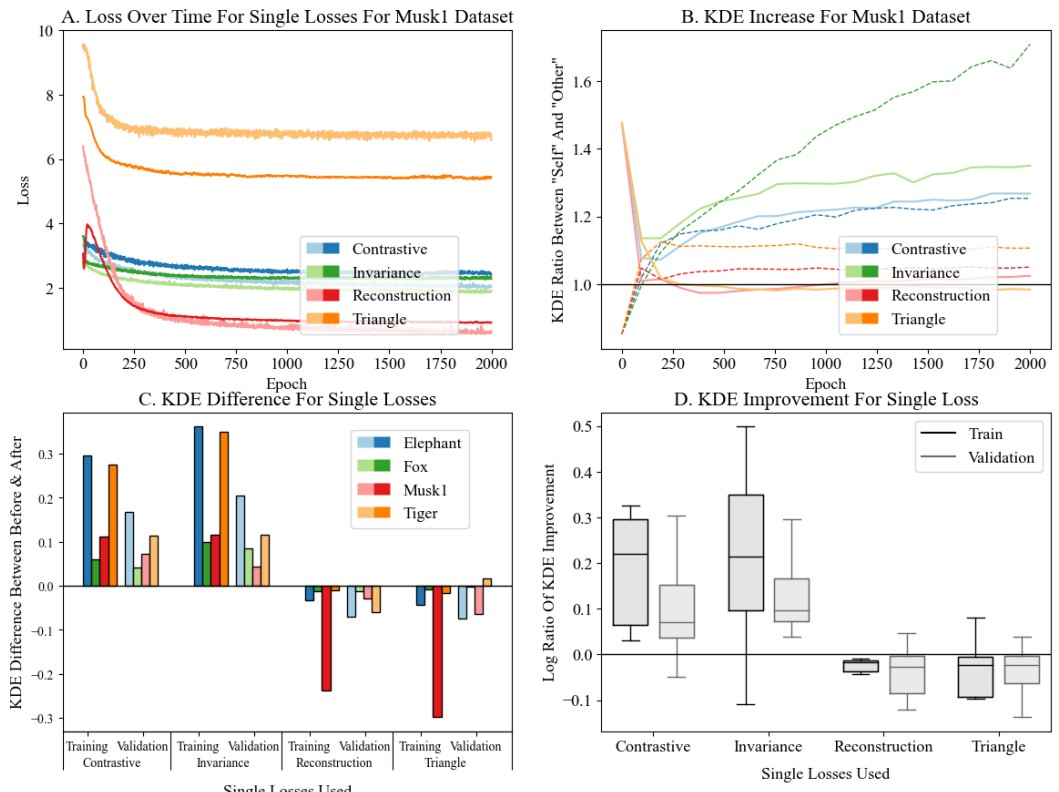

Figure 3: The four losses decay when training with a single loss at a time for the musk data **(a)**, each color accounting for a single loss, with the darker colored line for the training loss and the lighter for validation loss. An example of the KDE decay with the Musk data when training with a single loss at a time **(b)**, with the different colors for indicating the losses, where the light and dark trend lines exhibit the two different labels. The beginning-to-end ratio of KDE improvement $(KDEend - KDEbeginning)/KDEend$ for each dataset for each loss **(c)**, with the different colors indicating the losses, and the light and dark bars representing the two labels. Summarizing of the KDE improvement for each loss on all datasets **(d)**, with the two shades of gray denoting training and validation.

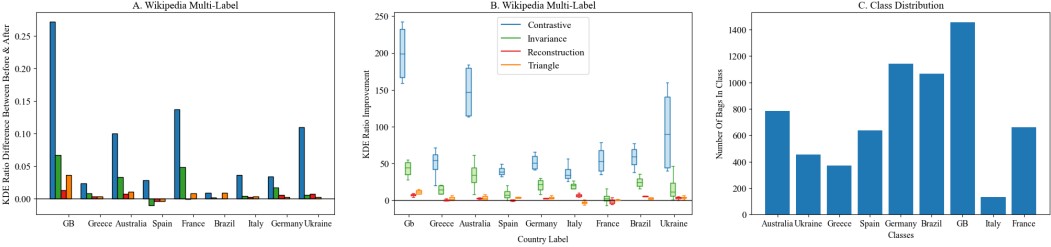

Figure 4: The beginning-to-end KDE improves $(KDEend - KDEbeginning)/KDEend$ for each of the four losses **(a)**. Each color accounts for a single loss. Distribution of the KDE improvement percentage for each loss for each label across all folds **(b)**, with the four colors corresponding to the four losses. The labels in the exhibited fold are almost balanced **(c)**, causing one label to improve more than others.

clustering loss with weights $0, 0.01, 0.1, 1, 10$. The clustering loss slightly improves the within-to-between class density (Fig 5a for wiki data and b for all other datasets). However, the difference

is non-significant (ANOVA on all methods and clustering loss weight). Note that the clustering is unstable, leading to large fluctuations in the loss and KDE. However, the clustering loss may be important for the representation, since a more dense cluster of bags aids with a clear distinction between the bags.

Given the importance of the contrastive and invariance loss and the failure of the reconstruction loss, we tested whether combining the losses can actually improve the within-to-between class density. Adding a reconstruction term to the loss consistently decreases the ratio. In contrast, the ratio of the contrastive to invariance loss is inconsequential, and both losses seem to be equivalent (Fig. 5c and d).

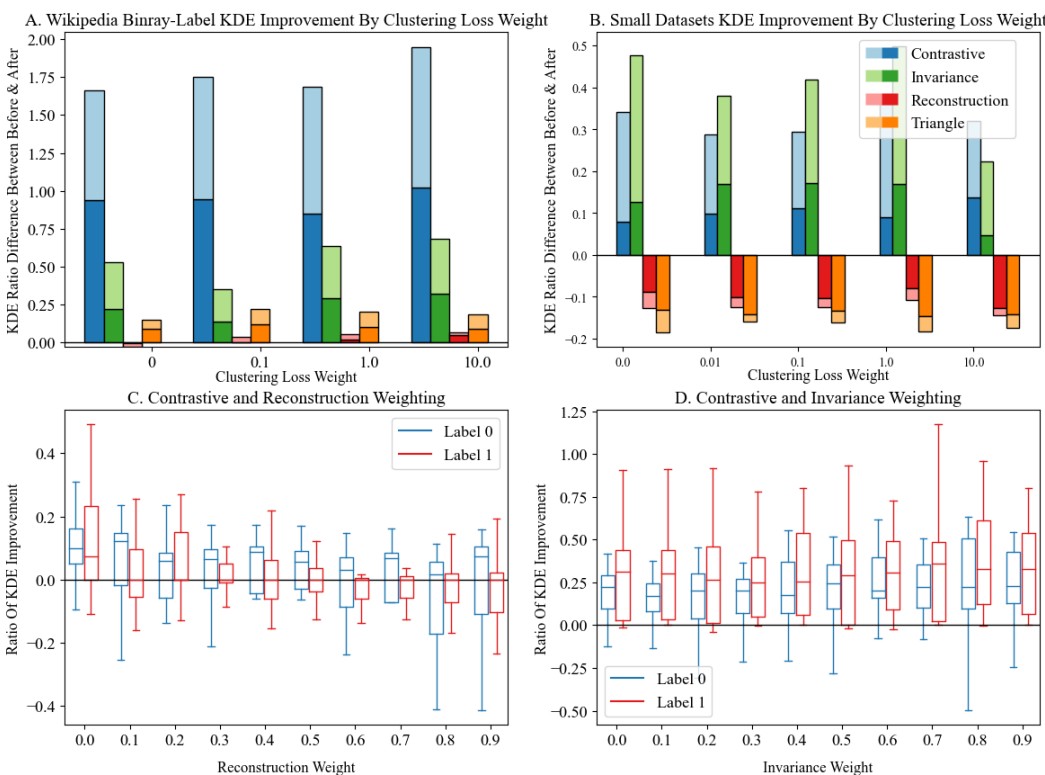

Figure 5: Clustering loss does not improve nor deteriorate. The models trained with clustering loss weighted with every other loss, for every weight of clustering loss while fixing the other loss weight to 1 for Wikipedia **(a)** and all other datasets **(b)**. The darker shade is for label 1, and the lighter is for label 0. The percentage of improvement $(KDEend - KDEbegining)/KDEend$ of the KDE before vs after training exhibits no distinct impact.Balancing between contrastive loss and other important losses reconstruction **(c)** and invariance **(d)** depends on the dataset.

## 5.6 CLASSIFICATION USING LEARNED METRIC

Finally, we tested whether the self-supervised loss also improved the classification accuracy. We used a simple KNN setup, where we defined the distance between each pair of bags using the learned embedding and the energy distance between bags. We then computed the classification accuracy. Following the self-supervised training. The test bags are classified using the majority vote of the 9 nearest neighbors for all datasets. We then computed the accuracy and increase in accuracy (vs the original instances values) of the KNN classification, either as a binary (average of each class vs all others), or multiclass (Fig. 6). As was the case for all other measures and datasets, in all cases, the contrastive and invariance loss improve the accuracy, while the triangle and reconstruction reduce the accuracy. The clustering loss has a limited effect.

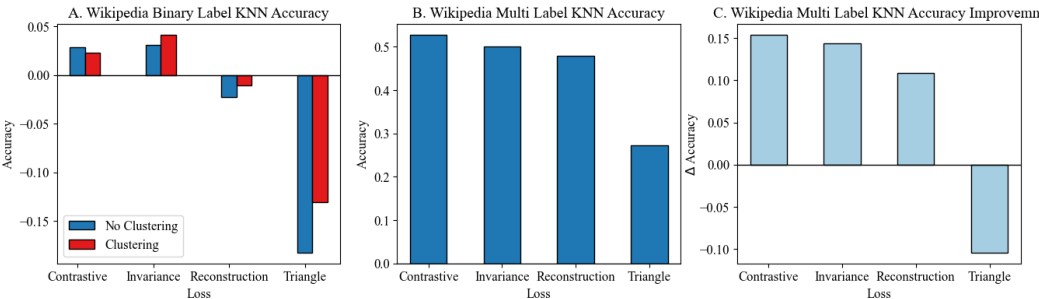

Figure 6: KNN classification accuracy. **(a)** for Wikipedia dataset for the binary label, with clustering loss and without. **(b)** KNN classification accuracy on the multi-label Wikipedia setup. **c** KNN classification accuracy improvement compared to KNN accuracy on original data

# 6    DISCUSSION

While MIL problems were extensively studied in supervised contexts, there are very limited solutions for distances between bags. We have here presented SUMIT a self-supervised approach to embed each instance to optimize the distance between bags. We used five types of distances: A reconstruction distance focused on reproducing the instance values as precisely as possible, a triangle distance that maintains the geometry of the projection, a clustering distance that ensures that the projection induces distinct clusters, and two bag-level self-supervised distances that minimize the distance within each bag. The first is a contrastive loss that compares the distance between and within bags, and the second distance ensures that the distance between bags is not sensitive to the number of instances sampled from each bag. Following the projection, we used an energy distance on the instances to define a bag distance.

We tested this distance definition by measuring the density of bags of different labels around bags of the same labels in standard datasets as well as a novel wiki-based dataset scrapped for the current task from the wiki pages of different cities. We have shown that while the instance-based distances (Triangles and reconstruction) do not increase, and sometimes decrease the density of bags of a given label around bags of the same label, the self-supervised distances increase this density. Thus, for the sake of an optimal distance, maintaining the bag information is more important than the precision of the instance reconstruction from the projection.

Adding the clustering distance to the bag level losses further improves the density of same-level bags, probably by ensuring more dense and separated projections.

SUMIT bridges the gap between bag-embedding methods where the entire bag is projected into a single vector in $\mathbb{R}^n$, and instance-based methods, where the focus is on the instances. Moreover, the combination of the clustering loss increases the typical self-density, but also the contrast between bags of different labels.

To the best of our knowledge, SUMIT is the first effort explicitly targeted at developing an instance-based distance between bags that is optimized at the bag level. Many choices in this analysis are arbitrary and can be changed. Those include among others the model used for the projection (currently a two-three-layer fully connected neural network), and the precise definition of the loss for each of the components (e.g. changing different types of self-supervised loss). However, from our experience, the precise details of either the encoder or the decoder, or the precise shape of the loss do not affect the qualitative results presented here.

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

## A    RELATED WORK AND APPLIED WORK

| Paper | Method | Distance |
|---|---|---|
| Citation-KNN | Hausdorff Distance | Hausdorff Distance |
| Multi-Instance Learning by Treating Instances As Non-I.I.D. Samples | MI-Graph | Graph Distance |
| Multiple Instance Learning with Bag Dissimilarities - MIND | Bag represented by vector of its dissimilarities to other bags which is the feature vector | Dissimilarity, not Distance |
| Contrastive Multiple Instance Learning: An Unsupervised Framework for Learning Slide-Level Representations of Whole Slide Histopathology Images without Labels | SIMCLR for instances, fuzing together with attention | Can be derived from encoding |
| Multi-instance clustering with applications to multi-instance prediction | Bag representations as a vector of distances to i medoids. | Hausdorff distance |
| Unsupervised Multiple-Instance Learning for Functional Profiling of Genomic Data | agglomerative or partition clustering and MIL's citationkNN | Measurement based of maximum pairs of instances with minimum MI |
| Unsupervised Multiple-Instance Learning For Instance Search | Finding Instance That Matches Bag Label | NA |
| **Applied Work** | | |
| Deep learning of feature representation with multiple instance learning for medical image analysis | Extracting features then classifying with them | NA |
| Dual attention multiple instances learning with an unsupervised complementary loss for COVID-19 screening | Contrastive loss is applied at the instance level to encode the similarity of features from the same patient against representative pooled patient features. | NA |

## B    WIKIPEDIA TOPICS

The Wikipedia dataset consists of 10 country-specific labels. For each country, we extracted pages for its cities, treating each city as a bag and the sections of its page as instances. The sections are represented as a bag of words, with sentences cleaned of stop words and other irrelevant words.

| Country | Number Of City Pages |
|---|---|
| GB | 5340 |
| Australia | 814 |
| Ukraine | 413 |
| France | 1149 |
| Greece | 347 |
| Spain | 481 |
| Italy | 754 |
| Germany | 987 |
| Brazil | 170 |

Table 2: Country Values Table

## C    NOTATIONS

We will use the following notations throughout the appendix:

1. A bag $\alpha$ of m instances will be marked $X_\alpha = x_1, x_2, ..., x_m$, where each $x_i$ is an instance of the bag.

2. Each instance $j$ is represented by a feature vector $x_j$ in a d-dimensional feature space (i.e., $x_j \in \mathcal{R}^d$).

3. An encoding of an instance will be marked as $\tilde{x}_i \in R^e$ in an e-dimensional encoding space, and for a bag $X_i$ with $n$ instances the encoding will be marked $\tilde{X}_i \in R^{eXn}$.

4. A decoding of an instance will be marked as $\hat{x}_i \in R^d$ in the original d-dimension feature space, and for a bag $X_i$ with $n$ instances the decoding will be marked $\hat{X}_i \in R^{dXn}$.

