# OpenReview forum: "Bag-level Self-supervised instance based distance in Multiple Instance Learning"
_ICLR.cc/2025/Conference — ICLR 2025 Conference Withdrawn Submission_

### Official Review · Reviewer_HypS · 2024-10-27

**Soundness:** 1
**Presentation:** 1
**Contribution:** 1
**Rating:** 3
**Confidence:** 4

**Summary:**

In this submission, authors focus on the distance metric in multiple instance learning. Specifically, they proposed a bag-level self-supervised instance based distance that maximizes the distinction between bags.

**Strengths:**

The topic focused in this paper, i.e., distance metric in multiple instance learning, is important for solving multiple instance learning problem. The authors propose SUMIT, an instance-embedding-based distance that maximizes the distinction between bags. Some experiments are conducted to show the effectiveness of the proposed method.

**Weaknesses:**

1. The motivation can be made clear. In the current version, it is unclear what problems exist in existing works and how the proposed method can address these problems.

2. The technical details of the proposed method is unclear. After reading Section 4, I am still unclear how does the proposed method work?

3. Authors should be aware that, this paper is to learn a distance metric.  This is not a new topic, and there have been many existing works, such as [a], [b], [c]. It is better to discuss them in related works or compare them in experiments.

4. The presentation quality should be greatly improved, including both writing skills and paper organization.

[a] Multi-instance Metric Learning, DOI: 10.1109/ICDM.2011.106

[b] A multi-task-based classification framework for multi-instance distance metric learning, DOI: 10.1016/j.neucom.2017.09.011

[c] Multiple Instance Metric Learning from Automatically Labeled Bags of Faces, DOI: 10.1007/978-3-642-15549-9_46

**Questions:**

It is better to rewrite the whole paper.

**Details Of Ethics Concerns:**

None.

---

### Official Review · Reviewer_Nuki · 2024-11-03

**Soundness:** 2
**Presentation:** 2
**Contribution:** 3
**Rating:** 5
**Confidence:** 3

**Summary:**

This study proposes SUMIT, which is an instance-embedding-based distance that maximizes the distinction between bags. SUMIT is optimized using five criteria: self-similarity within a bag, quality of instance reconstruction, robustness to sampling depth, conservation of triangle inequality, and separation of instances to clusters. The experiments present that the within-bag similarity loss is the most important for a bag-to-bag metric that best separates bags of similar classes. SUMIT bridges the gap between instance-level and bag-level approaches by keeping the embedding of all instances but ensuring their proximity within a bag.

**Strengths:**

1. The language of the manuscript is coherent and fluid.
2. The method proposed in this study offers novel techniques for the field of multi-instance learning, particularly concerning bag and instance distance metrics, which can also be applied in other areas such as metric learning.

**Weaknesses:**

1. The description of the innovations in this paper lacks clarity.
2. The experiments do not provide robust support for the claims made in this paper. In other words, although the paper presents several groups of experiments, the type is single, and the effectiveness of the proposed method is not well demonstrated.

**Questions:**

1.  What are the applications of the measurement methods proposed in this study, or rather, what are the benefits of such measurement methods in the context of multi-instance learning?
2.  What is the relationship between the five loss functions proposed? How does each loss function contribute to the effectiveness of the method?
3. Can the proposed method solve the sparsity or overlap between the instances of the bags? How did it do this?
4. Can the proposed method achieve the labeling at both instance level and bag label? How did it do this? Could you tell me more about its predictive performance? For an example, can you provide some comparison experiments?
5. SUMIT is applicable to the Wikipedia multi-class dataset. Was the experiment conducted by breaking it down into several binary classifications? What are the differences in applying the method in binary classification versus multi-class classification? Have there been considerations to apply it in the multi-label dataset? What new technical challenges could arise from this?

---

### Official Review · Reviewer_NRWV · 2024-11-03

**Soundness:** 2
**Presentation:** 1
**Contribution:** 1
**Rating:** 3
**Confidence:** 4

**Summary:**

This paper proposes a new Multi-Instance Learning (MIL) method called SUMIT (Self sUpervised MIL dIsTance), which aims to optimize the distance metric between bags through instance embedding. SUMIT optimizes through five criteria: self similarity of instances within the bag, quality of instance reconstruction, robustness to sampling depth, preservation of triangle inequalities, and separation from instances to clusters.

**Strengths:**

1. SUMIT is the first method to optimize instance distance at the bag level, which combines self-supervised learning with metric learning to generate metrics between bags, which is a novel research direction.

**Weaknesses:**

1.SUMIT has 5 criteria, but the weights between these 5 criteria and the role of each criterion are not specified.
2.Lack of comparative discussion with existing MIL methods.
3.Although this paper has some novelty, its presentation and contribution are far from sufficient.

**Questions:**

Please see the weaknesses above.

---

### Official Review · Reviewer_JQcN · 2024-11-04

**Soundness:** 2
**Presentation:** 2
**Contribution:** 1
**Rating:** 3
**Confidence:** 3

**Summary:**

This paper proposed an instance-embedding-based distance for the multiple instance learning with five different losses, reconstruction loss, contrastive loss, invariance loss, clustering, and triangle loss. The authors argue that many distances do not consider the distribution of instances for each bag and therefore propose to produce a metric for bags by combining self-supervised learning and metric learning. With the benchmarks, they show the proposed method , SUMIT, can bridge the gap between instance-level and bag-level.

**Strengths:**

Originality: this paper is related to the multiple instance learning by considering the distribution of instances in a bag. Specifically, they proposed to use five different losses and an energy distance for the embedding of each instance to ensure the distance between instances of different bags is larger than the instances in a bag. Instead of optimizing the distances between bags, the authors optimize the embedding of each instance based on five losses. The method sounds reasonable and the paper is clear for reading.

**Weaknesses:**

The paper has some experimental results on kernel density estimate for a toy dataset, and the ablation study for the different losses in the musk data, and the KDE improvement for each single loss, etc. However, the authors didn't show any comparison among the SOTA methods. Second, the paper claims it's a self-supervised instance method, however, it's hard to see the relationship in this paper. Third, this paper looks like a simple combination idea based on the energy distance [20], and different losses. Overall, it's not clear to see that why this problem is important and why consider those five different losses.

**Questions:**

Q1: Do the authors consider how to combine those losses and then optimize it? Do the authors have any discussion about any combination of any two or three losses on each dataset?
Q2: What is X' and Y' in the equation 10
Q3: It's not clear why using batch norm in encoder-decoder model and using layer norm for latent layer. It'd be better to have reference or explanation.
Q4: Are the five losses adopted to all the datasets? If yes, what's the results among the Wiki (text), Corel (image) and MUSK? A deep discussion for those different modalities will be good.
Q5: minor, Line 146, mentioned two types of dataset, looks there are three, including MUSK, Corel, and Wiki datasets.

---

### Note · Authors · 2024-11-19

I have read and agree with the venue's withdrawal policy on behalf of myself and my co-authors.